# A conserved MCM single-stranded DNA binding element is essential for replication initiation

**Clifford A Froelich[1][†], Sukhyun Kang[2][†], Leslie B Epling[1], Stephen P Bell[2]\*, Eric J Enemark[1]\***

[1]Department of Structural Biology, St Jude Children's Research Hospital, Memphis, United States; [2]Howard Hughes Medical Institute, Massachusetts Institute of Technology, Cambridge, United States

**Abstract** The ring-shaped MCM helicase is essential to all phases of DNA replication. The complex loads at replication origins as an inactive double-hexamer encircling duplex DNA. Helicase activation converts this species to two active single hexamers that encircle single-stranded DNA (ssDNA). The molecular details of MCM DNA interactions during these events are unknown. We determined the crystal structure of the *Pyrococcus furiosus* MCM N-terminal domain hexamer bound to ssDNA and define a conserved MCM-ssDNA binding motif (MSSB). Intriguingly, ssDNA binds the MCM ring interior perpendicular to the central channel with defined polarity. In eukaryotes, the MSSB is conserved in several Mcm2-7 subunits, and MSSB mutant combinations in *S. cerevisiae* Mcm2-7 are not viable. Mutant Mcm2-7 complexes assemble and are recruited to replication origins, but are defective in helicase loading and activation. Our findings identify an important MCM-ssDNA interaction and suggest it functions during helicase activation to select the strand for translocation.

**\*For correspondence:**
spbell@mit.edu (SPB);
eric.enemark@stjude.org (EJE)

[†]These authors contributed equally to this work

**Competing interests:** The authors declare that no competing interests exist.

**Reviewing editor**: Michael R Botchan, University of California, Berkeley, United States

## Introduction

Mcm proteins were first identified in yeast when mutations in their genes were defective for mini-chromosome maintenance (*Maiorano et al., 2006*). In eukaryotic cells, six related Mcm proteins (Mcm2-7) form a ring-shaped heterohexamer, the Mcm2-7 complex. Hexameric MCM rings act as the replicative DNA helicase (*Bochman and Schwacha, 2008*; *Ilves et al., 2010*), encircling the leading strand DNA template at the replication fork (*Fu et al., 2011*). Replication forks are established in a cell-cycle-regulated manner at specific regions of DNA called replication origins (*Bell and Dutta, 2002*). Mcm2-7 complexes are loaded onto double-stranded DNA at each replication origin by the Origin Recognition Complex (ORC), Cdc6, and Cdt1 (*Remus and Diffley, 2009*). Because replication origins are located far from the DNA ends, loading of Mcm2-7 hexamers such that they encircle double-stranded DNA requires opening of the Mcm2-7 ring. A 'gate' between the Mcm2 and Mcm5 subunits has been identified and is the likely site of ring opening and closing (*Bochman and Schwacha, 2007*, *2008*; *Costa et al., 2011*). After helicase loading, the two Mcm2-7 complexes encircle double-stranded DNA (dsDNA) as a head-to-head double hexamer (*Evrin et al., 2009*; *Remus et al., 2009*) that is inactive as a helicase.

Helicase activation requires substantial remodeling of the initially loaded Mcm2-7 double hexamer. The Dbf4-dependent Cdc7 kinase (DDK) and cyclin-dependent kinases (CDKs) drive recruitment of two Mcm2-7 activating proteins, Cdc45 and the tetrameric GINS complex (*Labib, 2010*). These proteins together stimulate the Mcm2-7 ATPase and helicase (*Ilves et al., 2010*) and with Mcm2-7 form the active replicative DNA helicase, the CMG complex (Cdc45-Mcm2-7-GINS) (*Moyer et al., 2006*;

**eLife digest** When DNA was first recognised to be a double helix, it was clear that this structure could easily explain how DNA could be replicated. Each strand was made of bases—represented by the letters 'A', 'C', 'G' and 'T'—and the two strands were held together by bonds between pairs of bases, one from each strand. Moreover, 'A' always paired with 'T', and 'C' always paired with 'G'. Therefore, if the two strands were separated, each could be used as a template to guide the synthesis of a new complementary strand and thus create two copies of the original double-stranded molecule. One of the first steps in this replication process involves a ring-shaped complex of six proteins, called an MCM helicase, separating the two strands.

To prepare for DNA replication, two MCM helicase rings wrap around the double-stranded DNA. Then, after the helicase has been activated, the bonds between the DNA base pairs break, and the two rings separate with one ring encircling each DNA strand. However, the details of the interactions between the helicase and the DNA during these events are not fully understood.

Now Froelich, Kang et al. have solved the three-dimensional structure of an MCM helicase ring—taken from a microbe originally found at deep ocean vents—on its own and also when bound to a short piece of single-stranded DNA. The helicase ring becomes more oval when the DNA binds to it. Moreover, rather than passing straight through the ring, the DNA wraps part of the way around the inside of the ring.

Specific amino acids—the building blocks of proteins—on the inside of the ring interact with the single-stranded DNA, and these amino acids are also found in MCM proteins in many other organisms. Furthermore, swapping these amino acids for different amino acids significantly reduced the ability of the ring to bind to single-stranded DNA, but its ability to bind to double-stranded DNA was only slightly affected. Engineering similar changes into the ring complexes of yeast cells was lethal, and the mutant complexes were less able to be loaded onto the DNA, or to be activated and separate the two strands ready for replication.

These insights into how helicases are loaded onto double-stranded DNA, and select one DNA strand to encircle, have improved our understanding of how DNA replication is initiated: a process that is vital for living things.

*Bochman and Schwacha, 2008*; *Ilves et al., 2010*). The initially loaded double-hexamer has the capacity to passively slide over dsDNA (*Evrin et al., 2009*; *Remus et al., 2009*), suggesting MCM DNA interactions are not fixed at this stage. Upon activation, the two Mcm2-7 helicases translocate independently (*Yardimci et al., 2010*) in a 3'→5' direction on the single-stranded leading strand DNA template (*Fu et al., 2011*). This transformation necessitates two structural changes in the initially loaded double-hexamer that are poorly understood: (i) the double-hexamer interface must be broken to allow independent replisome movement; (ii) the dsDNA at the origin must be melted and the lagging strand DNA template excluded from the central channel of each MCM hexamer. How Mcm2-7 retains one strand in its central channel while excluding the other during this transition is unknown.

Each Mcm subunit contains three domains. The N-terminal domain (MCM$_N$) possesses an OB (oligonucleotide/oligosaccharide binding)-fold and usually a zinc-binding motif (*Fletcher et al., 2003*). This domain mediates the head-to-head interaction of the two hexamers (*Gomez-Llorente et al., 2005*; *Evrin et al., 2009*; *Remus et al., 2009*). The second domain contains a conserved ATPase AAA+ fold (*Neuwald et al., 1999*), which binds and hydrolyzes ATP at subunit interfaces around the hexameric ring (*Schwacha and Bell, 2001*; *Davey et al., 2003*) and is required for DNA unwinding (*Bochman and Schwacha, 2008*; *Ilves et al., 2010*). A short domain at the C-terminus includes a helix-turn-helix fold (*Aravind and Koonin, 1999*), one of which (Mcm6) interacts with Cdt1 (*Wei et al., 2010*). MCM hexamers demonstrate a two-tiered ring architecture in electron microscopy studies with an N-terminal domain tier and an ATPase domain tier (*Chong et al., 2000*; *Pape et al., 2003*; *Gomez-Llorente et al., 2005*; *Costa et al., 2006*; *Bochman and Schwacha, 2007*; *Remus et al., 2009*; *Costa et al., 2011*). The MCM complexes of several archaeal organisms consist of six identical subunits and have provided powerful models to investigate the atomic details of MCM structure. Crystal structures have identified a consistent hexameric arrangement for MCM$_N$ of *Methanothermobacter thermautotrophicus* (*Mt*)

(*Fletcher et al., 2003*) and *Sulfolobus solfataricus* (*Sso*) (*Liu et al., 2008*) that correspond to the smaller tier observed by electron microscopy (*Remus et al., 2009*; *Costa et al., 2011*). Although no atomic structure has been determined for the complete archaeal or eukaryotic Mcm hexamer, hypothetical atomic models for full-length archaeal MCM hexamers have been generated by superimposition of six copies of a monomeric crystal structure of nearly full-length MCM onto the hexameric structure of *Mt*MCM$_N$ (*Brewster et al., 2008*; *Bae et al., 2009*).

Despite a growing understanding of the overall structure of the MCM complex, its multiple interactions with DNA during helicase loading, activation and elongation remain mysterious. Atomic structures of MCM bound to DNA have not been reported. Given the different forms of DNA that are bound to the MCM complex during the steps of the initiation pathway, the MCM proteins must transition between different DNA interactions during this process. To investigate the interactions after origin melting and how the MCM hexamer selectively encircles the leading strand template, we determined the crystal structure of the MCM$_N$ hexamer of *Pyrococcus furiosus* bound to ssDNA. We present an analysis of this the structure and biochemical and genetic characterizations of archaeal and *S. cerevisiae* proteins with mutations in the identified ssDNA binding region. These findings reveal two residues on the surface of the MCM OB-fold that are critical for MCM DNA-binding and contribute to multiple Mcm2-7 functions during replication initiation. Our findings support a model in which the identified MCM-ssDNA interactions contribute to the selection of the leading strand DNA template during helicase activation.

## Results

To elucidate how MCM interacts with ssDNA, we determined the crystal structure of the N-terminal domain of the *Pyrococcus furiosus* MCM (*Pf*MCM$_N$) protein in complex with homopolymeric (dT)$_{30}$ ssDNA (*Table 1*).

### MCM-ssDNA molecular architecture

The asymmetric unit of the crystal of *Pf*MCM$_N$:ssDNA contains two independent hexamers, each bound to ssDNA (*Figure 1*, *Figure 1—figure supplements 1,2*; *Video 1*). The subunits are referred to as A through F (hexamer 1) and G through L (hexamer 2). Like *Sso*MCM$_N$ (*Pucci et al., 2007*; *Liu et al., 2008*), *Pf*MCM$_N$ elutes as a monomer by size-exclusion chromatography (data not shown) but adopts a hexameric arrangement in the crystal structure. The structure is similar to those of *Mt*MCM$_N$ (*Fletcher et al., 2003*) and *Sso*MCM$_N$ (*Liu et al., 2008*) with three subdomains (*Figure 1—figure supplement 3*): a largely helical subdomain A; a Zn-binding subdomain B; and an OB-fold subdomain C. The central pore of the *Pf*MCM$_N$ hexameric ring is oval-shaped with a variable diameter around the ring reflecting a significant deviation from pure sixfold symmetry. The RMSD of the C-subdomain Cα-positions from the sixfold permutation is 3.03 Å and 1.45 Å for hexamers 1 and 2, respectively. In contrast, *Pf*MCM$_N$ without DNA bound is highly symmetric and shows minimal RMSD from sixfold symmetry (*Figure 1—figure supplements 4–6*, RMSD = 0.33 Å), indicating that DNA induces asymmetry in the MCM ring. The narrowest diameter of the channel is at the β-turn of the C-subdomain (*Figure 1—figure supplement 3*), consistent with previous structures of MCM$_N$ (*Fletcher et al., 2003*; *Liu et al., 2008*).

The ssDNA binds inside the central channel of the hexameric ring in an intriguing configuration. The ssDNA circles the interior of the *Pf*MCM$_N$ ring in a plane perpendicular to the central channel (*Figure 1*, *Figure 1—figure supplement 1*). This is in contrast to the ssDNA passing through the central channel, as observed in the structures of the nucleic acid complexes of the motor domains of the hexameric helicases E1 (*Enemark and Joshua-Tor, 2006*), Rho (*Thomsen and Berger, 2009*), and DnaB (*Itsathitphaisarn et al., 2012*). This distinction suggests that the newly identified MCM-ssDNA interactions might serve a function distinct from motor-driven helicase and translocase activities. The ssDNA binds to the MCM$_N$ OB-fold subdomain C at a region consistent with that of the prototype OB-fold protein SSB, but the ssDNA is oriented approximately perpendicular to that seen in SSB-ssDNA structures (*Figure 1—figure supplement 7*, *Raghunathan et al., 2000*; *Chan et al., 2009*). The ssDNA does not progress towards a specific end of the channel; therefore, the ssDNA does not have an assignable entry or exit direction from the ring. Instead, the ssDNA has a defined polarity relative to the MCM ring. When viewed from the C-terminal side of the complex (as shown in *Figure 1A*), the 5′ to 3′ direction of the bound ssDNA proceeds clockwise around the channel. This polarity is observed for both ssDNAs in each hexamer of the asymmetric unit.

**Table 1.** Data collection and refinement statistics

| | *Pf*MCM$_N$:dT$_{30}$ | *Pf*MCM$_N$ (no DNA) |
|---|---|---|
| Data collection | | |
| Space group | P2$_1$ | P2$_1$ |
| Cell dimensions | | |
| $a$, $b$, $c$ (Å) | 94.276, 113.397, 196.854 | 122.849, 103.064, 122.435 |
| α, β, γ (°) | 90, 101.354, 90 | 90, 119.85, 90 |
| Resolution (Å) | 50-3.20 (3.31–3.20) | 50-2.65 (2.74–2.65) |
| $R_{sym}$ | 0.109 (0.786) | 0.100 (0.569) |
| $I/\sigma I$ | 13.4 (1.64) | 16.3 (2.26) |
| Completeness (%) | 100 (100) | 98.8 (98.2) |
| Redundancy | 4.1 (4.1) | 3.7 (3.7) |
| Refinement | | |
| Resolution (Å) | 50-3.20 (3.29–3.20) | 50-2.65 (2.72–2.65) |
| No. reflections | 63497/3376 (4453/218) | 72376/3839 (5183/285) |
| $R_{work}$/$R_{free}$ | 0.257/0.294 (0.372/0.373) | 0.259/0.270 (0.484/0.502) |
| No. atoms | | |
| Protein | 24359 | 12258 |
| DNA | 584 | 0 |
| Zn$^{2+}$ | 12 | 6 |
| Water | 0 | 0 |
| *B*-factors | | |
| Protein | 129 | 78 |
| DNA | 179 | N/A |
| Zn$^{2+}$ | 204 | 145 |
| Water | N/A | N/A |
| R.m.s. deviations | | |
| Bond lengths (Å) | 0.008 | 0.011 |
| Bond angles (°) | 1.164 | 1.361 |

The structure reveals that the individual MCM subunits do not all simultaneously participate in ssDNA binding. In each hexamer, the bound nucleotides are not continuous but are separated into two stretches. Overall, two 7-mer stretches are observed in hexamer 1, and 11-mer and 4-mer stretches are observed in hexamer 2. The subunits that interact with DNA use a consistent binding mode with four nucleotides per subunit (**Figure 1**, **Figure 2**, **Figure 2—figure supplement 1**). The fourth nucleotide from the 5'-end of this binding mode is visible in the cases where it spans binding at adjacent subunits, but it is often disordered at the 3'-end of a ssDNA stretch. The four nucleotide per subunit binding increment contrasts with the motor domains of other hexameric helicases that bind either one (E1, **Enemark and Joshua-Tor, 2006**; Rho, **Thomsen and Berger, 2009**) or two (DnaB, **Itsathitphaisarn et al., 2012**) nucleotides per subunit and indicates that 24 nucleotides can bind if all the subunits simultaneously engage the ssDNA. The absence of ssDNA binding at some subunits is not due to insufficient DNA length because a 30-mer oligonucleotide was used for crystallization. The discontinuous DNA could result from the hexamer binding two separate 30-mer strands simultaneously or from the hexamer tightly binding one 30-mer ssDNA strand at two regions with the intervening nucleotides binding either weakly or not at all. We consider the latter to be more likely because binding of two parts of the same strand is anticipated to be cooperative.

The capacity of a subunit to bind ssDNA is determined by intersubunit distance (**Figure 1**, **Figure 2**, **Figure 2—figure supplement 1**). To compare the distance between different subunit pairs, we measured the distance between the R201 Cα atom of one subunit and the E127 Cα atom of the

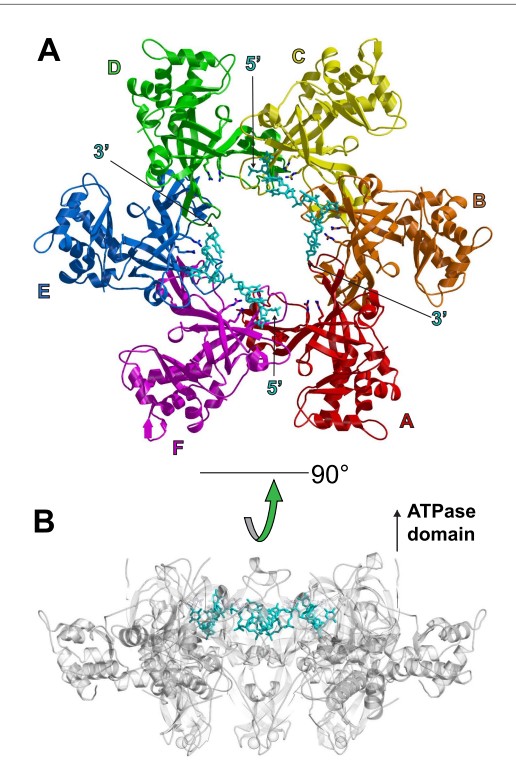

**Figure 1**. One crystallographically unique hexamer viewed parallel (A) and perpendicular (B) to the channel. The ssDNA is colored cyan. (**A**) Each subunit is uniquely colored and labeled. The side-chains of the two MSSB arginine residues that bind ssDNA are represented in stick. The Zn-binding domains project into the page. The ATPase domains, not present in the crystal structure, would project out of the page. (**B**) The protein is represented in transparent grey to highlight that the ssDNA runs perpendicular to the channel. The Zn-binding domains are at the bottom, and the ATPase domains would be located at the top.

The following figure supplements are available for figure 1:

**Figure supplement 1**. Views of the two hexamers of the crystallographic asymmetric unit parallel (A) and perpendicular (B) to the channel.

**Figure supplement 2**. Stereoimages of one ssDNA binding *Pf*MCM$_N$ subunit interface of each hexamer with Fo-Fc electron density calculated prior to including any DNA in the model.

**Figure supplement 3**. The ssDNA binds to the OB-fold subdomain.

**Figure supplement 4**. Crystal structure of *Pf*MCM$_N$ in the absence of DNA viewed parallel (A) and perpendicular (B) to the channel.

*Figure 1. Continued on next page*

counterclockwise subunit as viewed in *Figure 1* (Magenta arrow, *Figure 2*). DNA-binding is consistently observed at the first subunit if this distance is less than 7.5 Å, and it is not observed if this distance exceeds 8.4 Å. The interface between subunits J and K shows an intermediate (7.6 Å) distance, and the electron density between F202 (subunit J) and E127 (subunit K) is much weaker than at the interfaces where DNA has been modeled (*Figure 2—figure supplement 1*). The correlation of ssDNA binding with intersubunit configuration is conceptually similar to multi-subunit ATPase sites where different intersubunit configurations determine the ability to bind or hydrolyze ATP (*Abrahams et al., 1994*; *Enemark and Joshua-Tor, 2008*). In MCM$_N$, changes to the intersubunit configuration dictate binding to ssDNA.

## Conserved residues on the OB-fold bind ssDNA

The most significant interactions between *Pf*MCM$_N$ and ssDNA involve two adjacent arginines, R124 and R186, that project from the β-barrel of the OB-fold towards the ring interior (*Figures 1 and 2*). These residues interact with oxygen atoms of the sugars and bases of the ssDNA (*Figure 2*) and are highly conserved in other MCM proteins (*Figure 3*). We refer to this conserved region as the MCM Single-Stranded DNA Binding motif (MSSB). Interestingly, one thymidine base projects towards the β-barrel of the OB-fold (*Figure 2*) and makes two hydrogen bonds to main-chain atoms of one strand of the β-barrel. This base also sits at the subunit interface, between the side-chains of phenylalanine 202 of one subunit and glutamic acid 127 of the adjacent subunit. The β-turn residues R234 and K236 do not interact with ssDNA in the structure. The DNA-binding consists predominantly of interactions with the sugars and bases rather than the backbone phosphates. In contrast, the hexameric helicases E1 (*Enemark and Joshua-Tor, 2006*); Rho (*Thomsen and Berger, 2009*); and DnaB (*Itsathitphaisarn et al., 2012*) bind nucleic acid mainly through interactions with backbone phosphates.

We investigated the role of the identified residues in MCM DNA binding using mutational analysis and electrophoretic mobility shift assays. As expected, wild-type *Pf*MCM$_N$ binds single-stranded (*Figure 4*, $K_{half}$ = 6.8 μM) and double-stranded (*Figure 4—figure supplement 1*, $K_{half}$ = 7.0 μM) oligonucleotides. The arginine residues R124 and R186 make the most significant ssDNA interactions in the structure. R124A and R186A mutants each show a significant decrease in ssDNA binding

*Figure 1. Continued*

**Figure supplement 5**. Comparison of the crystal structures of *Pf*MCM$_N$ bound to ssDNA (left, in color) and in the absence of DNA (right, transparent grey). DOI: 10.7554/eLife.01993.009

**Figure supplement 6**. RMSD from sixfold symmetry for each crystallographic hexamer. DOI: 10.7554/eLife.01993.010

**Figure supplement 7**. Comparison of ssDNA binding by the *Pf*MCM$_N$ OB-fold subdomain C and by a prototypical OB-fold protein, SSB. DOI: 10.7554/eLife.01993.011

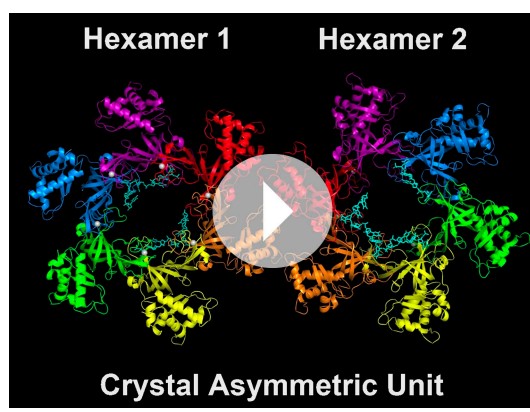

**Video 1**. Crystal structure details for PfMCMN:dT30. The video illustrates the asymmetric unit, which includes two MCM hexamers in a side-by-side orientation. Each subdomain is illustrated in Hexamer 1 to show that the ssDNA interacts with the OB-fold subdomain C. Finally, detailed views of the β-turn and the MCM Single-Stranded DNA binding motif (MSSB) are illustrated. DOI: 10.7554/eLife.01993.012

(7- and 6-fold reduction, respectively). Simultaneous mutation of both arginines showed even stronger defects (25-fold reduction), with no detectable ssDNA binding unless the protein concentration was increased dramatically (*Figure 4*). The K129A mutant is modestly defective in binding ssDNA (fourfold reduction, *Figure 4*). The individual R124A, R186A, and K129A mutants bind dsDNA with comparable affinity to wild-type (*Figure 4—figure supplement 1*). The R124A/R186A double mutant shows only modest defects in dsDNA binding (threefold reduction). Alanine mutants of other less-conserved residues did not significantly impair ssDNA- or dsDNA-binding. For example, consistent with the involvement of its main chain amide rather than its side chain in ssDNA binding, the β-turn K233A mutant does not significantly impair ssDNA binding. Similarly, the F202 side-chain interacts with a thymidine base, but it is off-set from an ideal stacking interaction (*Figure 2*). The corresponding F202A mutant is not impaired in ssDNA binding and is not conserved as aromatic in other Mcm proteins (*Figure 3*).

## Corresponding yeast MCM2-7 mutants are defective in vivo

In *S. cerevisiae* (*Sc*), the *Pf*MCM R124 and R186 amino acids within the MSSB motif are both conserved as arginine or lysine in Mcm4, Mcm6 and Mcm7 whereas Mcm2, Mcm3 and Mcm5 show a positively charged residue at only one of the two sites (*Figure 3*). To test the role of the MSSB motif in *S. cerevisiae* DNA replication, we constructed double-alanine mutants in *ScMCM4* (*mcm4-R334A/K398A = mcm4D*), *ScMCM6* (*mcm6-R296A/R360A = mcm6D*) and *ScMCM7* (*mcm7-R247A/K314A = mcm7D*) as these subunits showed the most similarity to *Pf*MCM in the MSSB.

We tested the ability of these mutations to replace the corresponding wild-type Mcm subunit in *S. cerevisiae* cells. When present as the only mutant Mcm subunit in the cell, mutations in the *ScMCM4*, *ScMCM6* or *ScMCM7* MSSB complemented deletion of the corresponding gene (*Figure 5A*, *Figure 5—figure supplement 1*). Because the DNA binding defects observed for the mutant *Pf*MCM complexes altered all six subunits, we tested the ability of pairwise combinations of the *ScMCM* MSSB mutations to function in place of their wild-type counterparts. In contrast to the single mutations, all three double-mutant combinations did not support cell division. The dramatic phenotypic difference between the double and single mutations may be due to a requirement for two adjacent subunits to create a productive ssDNA interaction. Because the Mcm4, 6 and 7 subunits are adjacent to one another in the Mcm2-7 complex, each pairwise combination would be expected to interrupt at least three possible subunit pairs for binding (e.g., the Mcm4/6 double mutant would interfere with Mcm2/6, Mcm6/4 and Mcm4/7 subunit pairs for ssDNA binding).

## *S. cerevisiae* MCM2-7 mutants exhibit helicase loading and replication initiation defects

To define further the molecular defects of the mutant *S. cerevisiae* Mcm2-7 complexes, we purified Mcm2-7 complexes containing the lethal double mutants (Mcm4D/6D, Mcm4D/7D and Mcm6D/7D) along with the associated Cdt1 protein. We also purified the Mcm4/6/7 triple mutant (Mcm4D/6D/7D) with associated Cdt1. After purification, all of the mutant complexes showed similar subunit composition

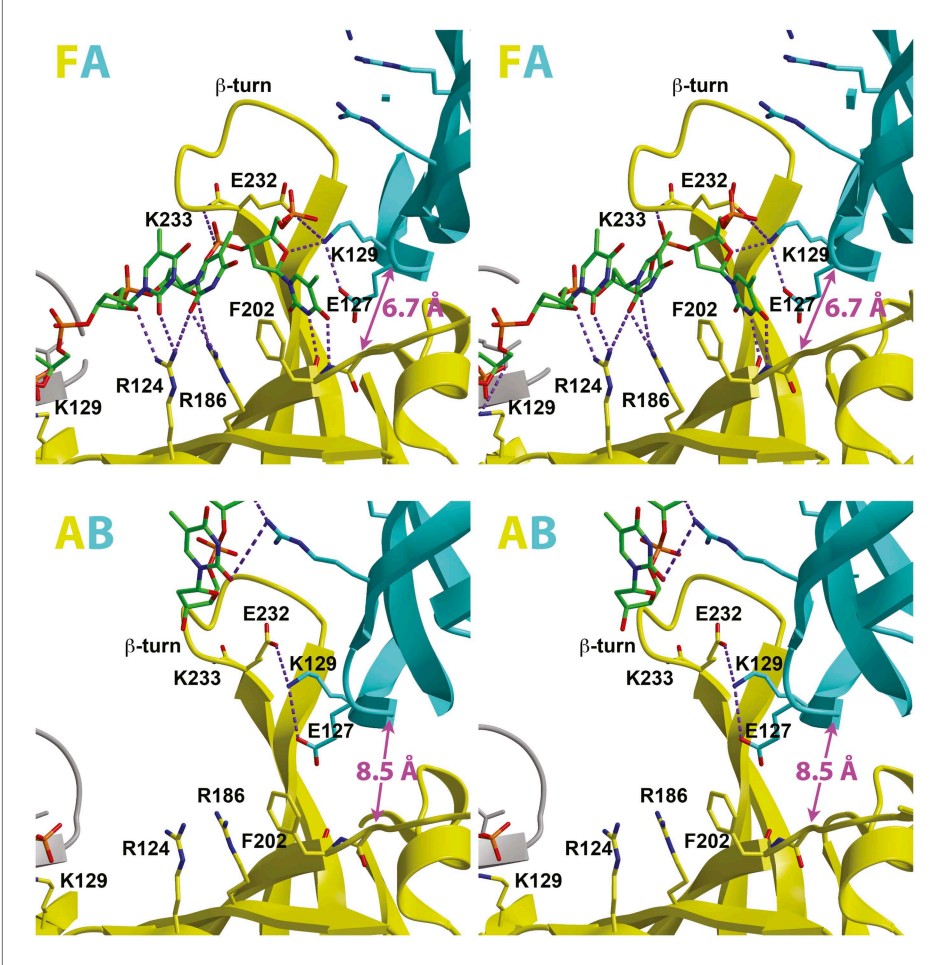

**Figure 2**. Stereoviews of the protein-DNA interaction details for two subunit interfaces. The binding predominantly involves residues on the face of the OB-fold of one subunit, yellow, including an interaction between a thymidine base and main-chain atoms of the β-strand. This thymidine is sandwiched between F202 of one subunit and E127 of the adjacent subunit in cyan. Lysine 129 of the neighboring subunit (cyan) interacts with both the DNA and the yellow subunit. The specific interfaces depicted are (top) between chains F (yellow) and A (cyan) and (bottom) between chains A (yellow) and B (cyan). The structural details of DNA-binding appear highly similar at the other interfaces where DNA is observed (see *Figure 2—figure supplement 1*). The main interactions involve R124 and R186. The presence of ssDNA correlates with the proximity of the two subunits as defined by the distance between the R201 Cα and E127 Cα positions (magenta arrow).

The following figure supplements are available for figure 2:

**Figure supplement 1**. 12 stereoimages of the *Pf*MCM interfaces sorted by intersubunit distance to emphasize the correlation with DNA-binding.

and migration in gel filtration columns as wild-type Mcm2-7/Cdt1 (*Figure 5—figure supplement 2*). Thus, these mutations do not inhibit the initial assembly of the Mcm2-7/Cdt1 complex.

We tested each of the mutant complexes for their ability to be loaded onto origin DNA using a reconstituted helicase-loading assay (*Evrin et al., 2009*; *Remus et al., 2009*; *Figure 5B*). To ensure that all of the Mcm2-7 hexamers retained on the DNA were loaded, we washed the final DNA-associated proteins with high salt. This treatment removes all of the helicase loading proteins (ORC, Cdc6 and Cdt1) from the DNA but leaves loaded Mcm2-7 complexes (*Figure 5B*, top panel) (*Randell et al., 2006*). Wild-type protein showed robust, Cdc6-dependent loading onto origin DNA. In contrast, each of the double mutant Mcm2-7 complexes showed reduced Mcm2-7 loading. The Mcm4D/6D and Mcm6D/7D complexes showed only modest defects (less than ~ two-fold, *Figure 5—figure supplement 2*).

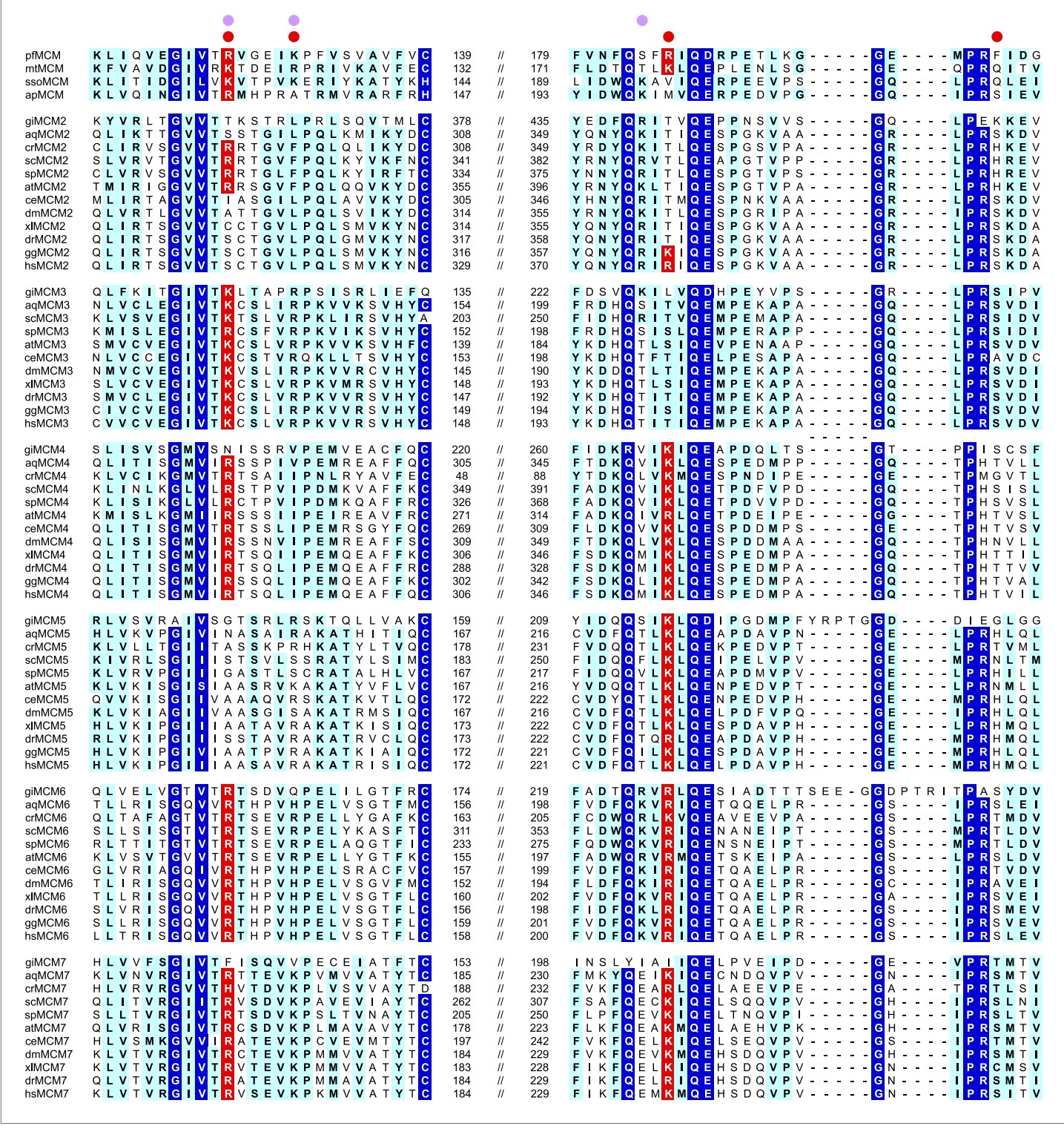

**Figure 3**. MCM family-specific sequence-alignment in the regions where the strongest interactions with ssDNA are observed. Globally conserved residues are shaded dark blue, and family-specific conserved residues are shaded light blue. Residues identified to participate in DNA-binding from our structure (red dot) and prior work (*Pucci et al., 2004*) (lavendar dot) are noted above the sequences. Conserved residue positions for ssDNA binding are shaded red and correspond to R124 and R186 in *Pf*MCM (*Figure 2*). pf = *Pyrococcus furiosus*; mt = *Methanothermobacter thermautotrophicus*; sso = *Sulfolobus solfataricus*; ap = *Aeropyrum pernix*; gi = *Giardia lamblia*; aq = *Amphimedon queenslandica*; cr = *Chlamydomonas reinhardtii*; sc = *Saccharomyces cerevisiae*; sp = *Schizosaccharomyces pombe*; at = *Arabidopsis thaliana*; ce = *Caenorhabditis elegans*; dm = *Drosophila melanogaster*; xl = *Xenopus laevis*; dr = *Danio rerio*; gg = *Gallus gallus*; hs = *Homo sapiens*.

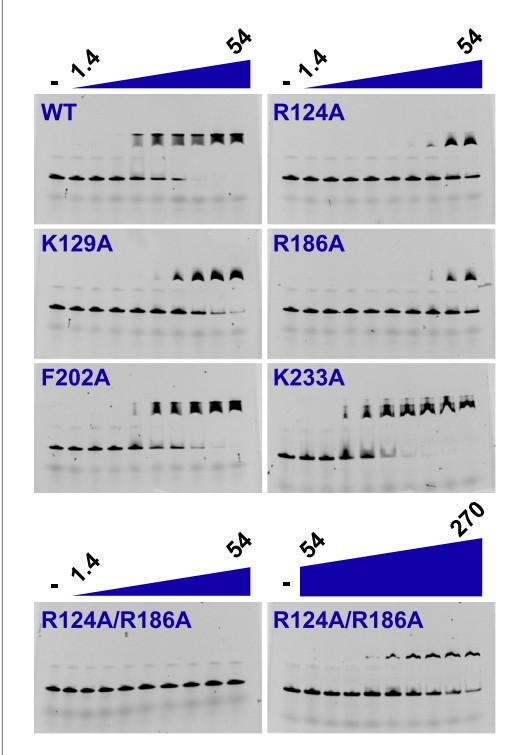

**Figure 4**. Electrophoretic mobility shift of 40-mer oligo-dT in the presence of *Pf*MCM$_N$. The ssDNA, 160 nM with a 5′-fluorescein-label, was titrated with increasing concentrations (1.4, 2, 2.7, 6.8, 13.5, 20.3, 27, 40.5, 54 μM) of *Pf*MCM$_N$. The lane marked '−' is loaded with control sample lacking protein. Mutation of residues R124 and R186 significantly impairs binding to ssDNA. The R124A/R186A double mutant was titrated with larger concentrations (54, 81, 108, 135, 162, 189, 216, 243, 270 μM) of *Pf*MCM$_N$ in order to detect binding.

The following figure supplements are available for figure 4:

**Figure supplement 1**. Electrophoretic mobility shift assay of a 26-mer dsDNA substrate in the presence of *Pf*MCM$_N$.

The Mcm4D/7D complex showed a stronger defect (~10-fold), and the Mcm4/6/7 triple mutant showed the most severe defect in helicase loading (~20-fold reduction, *Figure 5—figure supplement 2*).

To establish at what step in the helicase loading process these defects occurred, we studied the initial recruitment of the complexes to origin DNA. To this end, we replaced ATP with the poorly hydrolyzable ATP-γS in the assay. In the presence of ATPγS, all of the proteins required for helicase loading are recruited to the origin, but no loading occurs (*Randell et al., 2006*). Under these conditions, we observed a similar pattern of Mcm2-7/Cdt1 and ORC association for wild-type and the mutant Mcm2-7 complexes (*Figure 5B*, middle panel, *Figure 5—figure supplement 2*). Thus, mutating two or three MSSB motifs did not alter the initial recruitment of the Mcm2-7/Cdt1 complex to the origin DNA. We also examined the DNA-associated proteins when ATP-containing reactions were washed with low-salt (*Figure 5B*, bottom panel, *Figure 5—figure supplement 2*), a condition that retains helicase-loading proteins on DNA. Under these conditions, the mutant complexes showed a similar pattern of reduced Mcm2-7 DNA association as seen for the high-salt wash experiments. Cdt1 was not retained on the DNA under these conditions for mutant Mcm2-7 complexes, indicating that the MSSB mutations did not interfere with the release of Cdt1 from the Mcm2-7 complex during loading. Together, these data indicate that the loading defect for these Mcm2-7 mutants occurs after their initial recruitment to origin DNA but before the establishment of the ORC-independent association of Mcm2-7 with origin DNA.

We looked for additional replication initiation defects for the Mcm2-7 mutants that showed detectable loading using a modified in vitro replication assay that recapitulates origin-dependent DNA replication initiation and elongation (*Heller*

*et al., 2011*). In contrast to our original studies, helicase loading in these assays was performed using purified proteins. In addition to measuring new DNA synthesis, we monitored association of Mcm2-7, the helicase activation proteins Cdc45 and GINS and the ssDNA binding protein, RPA, with the origin DNA during the reaction. The analysis of protein associations provided insights into the step during replication initiation during which the mutant Mcm2-7 complexes fail. Consistent with their inability to support cell growth, none of the mutant complexes supported significant DNA synthesis (*Figure 6*). Analysis of FLAG-Mcm3 DNA association showed that, as in the loading assays, the Mcm4D/6D and Mcm6D/7D complexes are retained on the DNA more strongly than the Mcm4D/7D complex. Cdc45 association mirrored the level of FLAG-Mcm3 association with the DNA, suggesting Cdc45 recruitment is independent of the MSSB (*Figure 6—figure supplement 1*). In contrast, all of the Mcm2-7 double mutants showed similarly strong defects (≥10-fold) in both GINS and RPA DNA association. In the case of Mcm4D/7D mutant, the DNA replication, GINS and RPA DNA association defects are consistent with its helicase-loading defect. In contrast, for Mcm4D/6D and Mcm6D/7D, the extent

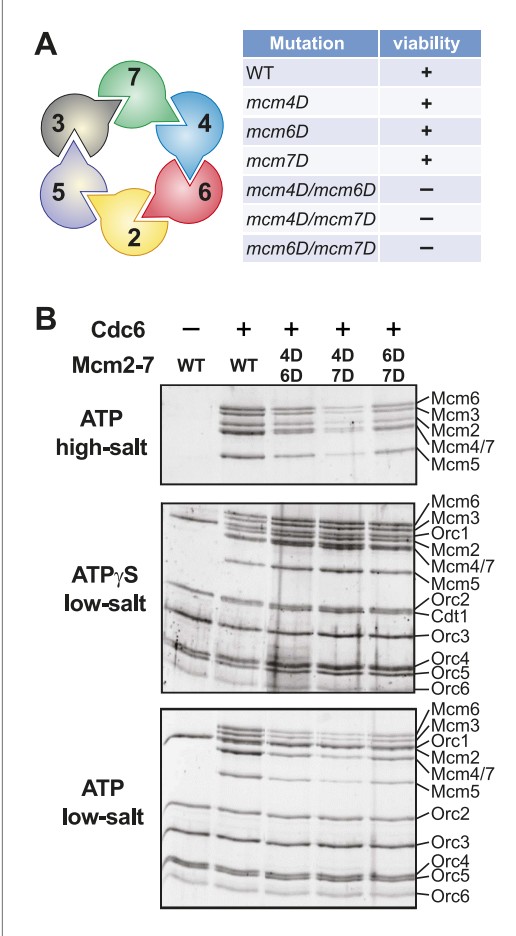

**Figure 5**. Mutation of two MSSB motifs is lethal and causes helicase loading defects. (**A**) Mutation of two Mcm4, 6, 7 MSSB motifs is lethal. Subunit arrangement in the Mcm2-7 ring viewed from the C-terminal side. The Mcm4, 6, and 7 subunits are adjacent to each other across from the Mcm2/5 gate. All pairwise combinations of the Mcm4, 6 and 7 MSSB mutants are lethal whereas the individual MSSB mutants are viable. (**B**) Helicase loading with the indicated MSSB double mutant Mcm2-7 complexes. Three forms of the assay are shown: following a high-salt wash to monitor completion of loading (top panel); in the presence of ATPγS instead of ATP to monitor the initial association of the helicase and all of the helicase loading proteins (ORC, Cdc6 and Cdt1, middle panel); and with ATP following a low salt-wash, allowing bound helicase loading proteins to be maintained (bottom panel). All loading was dependent on Cdc6 and proteins are detected after SDS-PAGE and fluorescent protein staining.

The following figure supplements are available for figure 5:

**Figure supplement 1**. All pairwise combinations of *mcm4D*, *mcm6D* and *mcm7D* mutants were not viable.

**Figure supplement 2**. Comparison of wild-type and MSSB double- and triple-mutant Mcm2-7/Cdt1 complexes.

of helicase loading and Cdc45 DNA association is distinct from the much larger losses in GINS and RPA DNA association and DNA replication (*Figure 6—figure supplement 1*). These data strongly suggest that an inability to recruit or maintain GINS and/or RPA is responsible for the replication defects exhibited by these mutants. Because RPA DNA binding is a readout for ssDNA formation and GINS is required to activate the Mcm2-7 helicase, both of these defects indicate that the Mcm4/6 and Mcm6/7 MSSB mutants are defective for helicase activation.

## Discussion

Here we show how the *Pf*MCM N-terminal domain interacts with single-stranded DNA and identify a critical set of interacting residues that we define as the MSSB. These residues are important for binding ssDNA and, to a lesser extent, dsDNA. A DNA-binding role for positively charged residues in this region is consistent with previous mutational analysis of *Sso*MCM showing that K129A (equivalent to *Pf*MCM R124) displays very little binding to ssDNA, blunt duplex DNA, and bubble-DNA substrates (*Pucci et al., 2004*). Although a positive residue equivalent to *Pf*MCM R186 is not conserved in *Sso*MCM, mutation of an adjacent residue, K194A also displays very little binding to these DNA substrates (*Pucci et al., 2004*). As previously noted in overall sequence comparisons (*Pucci et al., 2004*), residues in the MSSB motif are conserved in specific families in eukaryotic Mcm2-7. Importantly, we show that conserved residues within this motif are critical for *S. cerevisiae* cell division and multiple Mcm2-7 functions during replication initiation.

Biochemical analysis of the *S. cerevisiae* mutant complexes reveals multiple defects during replication initiation. Two mutant complexes (Mcm4D/7D and Mcm4D/6D/7D) show strong defects in Mcm2-7 loading. This is unexpected because Mcm2-7 proteins are loaded around dsDNA and there is no evidence for ssDNA at this stage of replication (*Evrin et al., 2009*; *Remus et al., 2009*). It is possible that one or more MSSB motifs interact with dsDNA prior to ssDNA formation at the origin and that these interactions stabilize loaded Mcm2-7. This would be consistent with the dsDNA binding defects observed for the *Pf*MCM_N R124A/R186A double mutant (*Figure 4—figure supplement 1*) and also the (R124-equivalent) K129A mutant of *Sso*MCM. Alternatively, elimination of positive charges in the central channel could alter the opening and closing of the Mcm2-7 ring. The abundance of positive charges in the Mcm2-7 ring could predispose the ring to remain

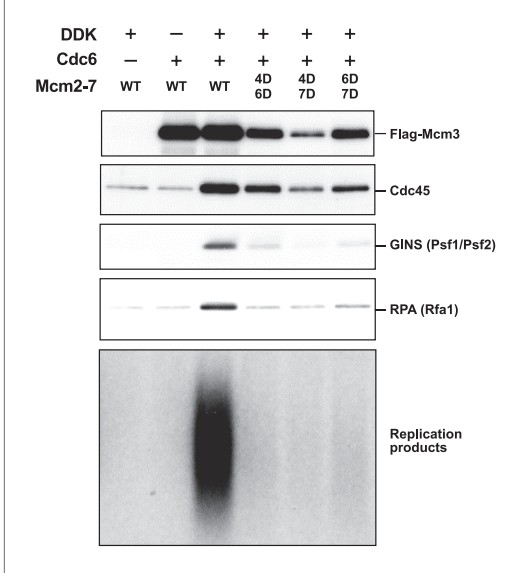

**Figure 6**. The Mcm2-7 MSSB double mutants are severely defective for in vitro DNA replication. Proteins associated with the DNA template during DNA replication were analyzed by immunoblotting (top panels) and radiolabeled DNA replication products were analyzed by alkaline agarose electrophoresis (bottom panel). All of the mutants are strongly defective for DNA replication and GINS and RPA DNA template association relative to wild-type Mcm2-7. The levels of Cdc45 and Mcm2-7 (FLAG-Mcm3) association reflected the levels of helicase loading by the same MSSB double mutant Mcm2-7 complexes. Quantitation of these data is shown in *Figure 6—figure supplement 1*.

The following figure supplements are available for figure 6:

**Figure supplement 1**. Quantitation of DNA template association of Mcm3, Cdc45, GINS and RPA and DNA replication products for the Mcm2-7 mutants relative to wild-type.

open prior to DNA binding. Encircling dsDNA could neutralize the repulsion and favor ring closing. It is possible that a reduction in positive charge in the mutant complexes leads to the Mcm2-7 ring spending more time in the closed state, inhibiting entry of the dsDNA during loading. Analogously, the reduced positive charge of the MSSB mutants could destabilize ring closure around dsDNA during loading. Consistent with this model, the Mcm2-7 complex appears as a cracked-ring in solution (*Costa et al., 2011*). As we observe, both scenarios predict that the strongest loading defects would be observed for the Mcm4D/6D/7D mutant that eliminates the greatest number of positive charges. Among the double mutants, the strongest loading defect is observed when the Mcm4 and Mcm7 subunits are mutated, which are across from the Mcm2/5 gate and could influence opening and closing more than other subunits.

Several lines of evidence suggest that the MCM-ssDNA interactions that we have identified have a role during dsDNA melting. First, the MCM-ssDNA interactions identified in our structure predominantly involve the sugars and bases of the ssDNA, ideally suited to bind and shield one strand from its complement during melting. Also consistent with a role in dsDNA melting, the Mcm2-7 MSSB mutant complexes showed strong defects in events linked to helicase activation. The MSSB mutations did not alter Cdc45 recruitment, consistent with the observation that this event can occur in G1 phase prior to ssDNA formation (*Aparicio et al., 1999*; *Heller et al., 2011*; *Tanaka et al., 2011*). In contrast, the levels of GINS and RPA DNA association by each of the MSSB mutant complexes were strongly defective. The defect in RPA DNA binding is almost certainly due to reduced ssDNA generation by the mutant complexes. The reduction in DNA-associated GINS could be the result of a defect in recruitment or retention of GINS. Unlike Cdc45, GINS recruitment does not occur until entry into S phase (*Kanemaki et al., 2003*; *Takayama et al., 2003*) and, therefore, could require ssDNA formation. Alternatively, it is possible that the defect in ssDNA binding prevents the CMG complex from attaining a particular DNA binding state and this destabilizes GINS binding.

Interactions between the MSSB and ssDNA could also occur during elongation. Consistent with a role for the MSSB in unwinding, the *Sso*MCM K129A mutant (*Pf*MCM R124 equivalent) is defective for helicase activity (*Pucci et al., 2004*). Although the MCM ATPase domain alone is sufficient to produce unwinding activity in *Sso*MCM (*Barry et al., 2007*; *Pucci et al., 2007*) and in *Aeropyrum pernix* MCM (*Atanassova and Grainge, 2008*), unwinding displays greater processivity in the presence of the N-terminal domain for *Sso*MCM (*Barry et al., 2007*). Thus, although the N-terminal domain and the residues of the MSSB are not intrinsically required to produce an unwinding activity, the N-terminal domain can regulate and enhance MCM unwinding activity (*Barry et al., 2007*). The positively charged residues of the MSSB could help maintain a closed MCM ring as described above for loading, and thus contribute to the enhanced processivity afforded by the N-terminal domain. It is also possible that ssDNA binding by the MSSB has a more direct impact on DNA unwinding. For example, the directional

ssDNA:MSSB interactions observed here could influence the polarity of unwinding either during initiation (see below) or elongation. To permit the ssDNA:MCM$_N$ interactions that we observe, the ssDNA would need to alter its trajectory as it passes through the MCM central channel. Alternatively, the MSSB could bind ssDNA differently during unwinding. An interesting possibility is that during elongation the MCM OB-fold binds ssDNA similar to the OB-fold prototype SSB (*Raghunathan et al., 2000*; *Chan et al., 2009*). This mode of binding would place the ssDNA approximately parallel to the central channel (*Figure 1—figure supplement 7*), a position consistent with the expected ssDNA trajectory during unwinding. Different modes of interaction between the MSSB and ssDNA could be modulated by the AAA+ domain of MCM and a conserved 'allosteric communication loop' (ACL, *Sakakibara et al., 2008*, *Barry et al., 2009*) that projects from the N-terminal domain towards the anticipated position of the ATPase domain. The ACL directly follows the β-strand that contains the second positively charged MSSB residue (*Pf*MCM residue R186) and thus could couple the MSSB to the ATPase domains.

The polarity of ssDNA bound to MCM$_N$ observed in our structure has important implications for the transition between MCM dsDNA and ssDNA binding. In the view shown in *Figure 7A*, the AAA+ motors are located above the MCM$_N$ domain, and the corresponding Mcm2-7 subunits occur clockwise in the order Mcm5, 3, 7, 4, 6, 2. Given that the Mcm2-7 complex is initially loaded around dsDNA, only one of the two strands of dsDNA can easily attain the 5'→3' coplanar clockwise configuration observed in our structure: the DNA strand that passes from the C- to N-terminus of the MCM complex in a 5'→3' direction (*Figure 7A*). Intriguingly, this strand corresponds to the leading strand DNA template that is encircled by the MCM complex during translocation/DNA unwinding. For the opposite strand to interact with the MCM$_N$ with the observed polarity, it would either need to pass through the other strand or dramatically re-orient. Thus, if ssDNA is formed within the MCM ring during origin melting (see below), our structure predicts that MCM$_N$ would preferentially bind to the translocating strand (i.e., the leading strand DNA template). Consistent with this model, the 3'→5' helicase polarity of *Sso*MCM is only observed when the N-terminal domain is present, implicating this domain in substrate selection (*Barry et al., 2007*).

The MCM helicase is conceptually similar to the Rho hexameric helicase because both possess an N-terminal OB-fold linked to a C-terminal ATPase. This analogy further supports a role for the MCM OB-fold during helicase activation prior to unwinding. The crystal structure of Rho with RNA bound at the OB-fold (*Skordalakes and Berger, 2003*) suggests that 70–80 nucleotides of RNA would adopt a circular path around the ring (*Skordalakes and Berger, 2003*) that is roughly perpendicular to the hexameric channel. This arrangement is conceptually similar to our *Pf*MCM$_N$:ssDNA structure. The Rho OB-fold is believed to bind RNA and facilitate encircling of single-stranded RNA during ring closure by the ATPase domains (*Skordalakes and Berger, 2003*), a prerequisite for establishing an activated helicase. Subsequently, the proposed unwinding mechanism for Rho exclusively involves distinct interactions between the ATPase motor domain and RNA (*Thomsen and Berger, 2009*). The MCM N-terminal domain may also function to enable the ATPase domains to select and encircle one strand of DNA during ring closure. A key difference between MCM and Rho is that the Rho helicase ring is loaded on a species that is already single-stranded, whereas the MCM hexamer is first loaded onto double-stranded DNA that must somehow be converted to single-stranded DNA (*Evrin et al., 2009*; *Remus et al., 2009*).

Combining the features of eukaryotic MCMs with our new structural information, we suggest the following model for MSSB function during helicase activation. After helicase loading, we propose that DNA melting is initiated by activating the ATPase domains of the double-hexamer to pump dsDNA from the C-terminal lobe side towards the double-hexamer interface (*Figure 7B*; *Video 2*). This is consistent with the known direction of MCM DNA translocation (*McGeoch et al., 2005*) as well as observations that Mcm complexes can translocate on dsDNA (*Kaplan et al., 2003*). As additional nucleotides of DNA are forced to occupy the same distance along the DNA helical axis, a B-form structure can no longer be maintained. We predict that the DNA strands would be forced apart at the site where the diameter of the MCM central channel is largest. Intriguingly, the MSSB is on the surface of the largest diameter of the MCM$_N$ central channel (*Figure 1—figure supplement 3*), putting the MSSB in a prime position to bind the leading strand ssDNA upon DNA melting. The channel diameter elsewhere in the MCM$_N$ is too narrow to permit B-form DNA strands to separate. Such melting activity requires that the two hexamers are anchored to one another because the two hexamers would otherwise simply translocate away from one another without melting the DNA. Further dsDNA pumping after the volume around the MSSB has been filled would require the MCM ring to open and allow the unbound lagging strand DNA template to exit (*Figure 7A*). The presumed exit site would be through the Mcm2/5 gate (*Bochman and Schwacha, 2007*; *Costa et al., 2011*). Intriguingly, Mcm2

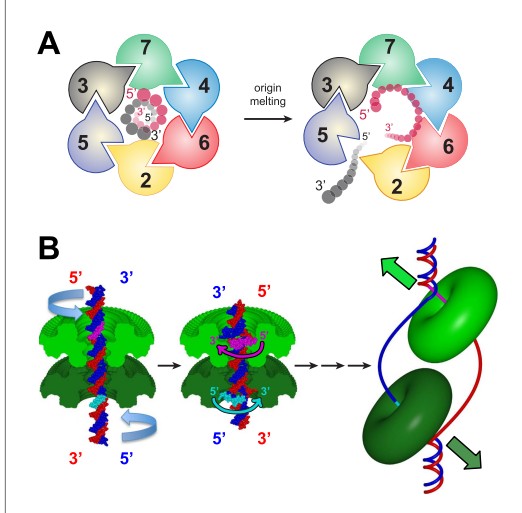

**Figure 7**. A model for MSSB-dependent selection of the translocating DNA strand during helicase activation. (**A**) The defined polarity of ssDNA binding by the $MCM_N$ would preferentially bind the leading-strand DNA template. The Mcm2-7 complex N-terminus is shown from the C-terminal side of the complex. This is the side where DNA is expected to enter during translocation. Duplex DNA is first encircled by the ring (left). Only the red strand can readily attain the 5′→3′ clockwise polarity observed in the crystal structure. This strand passes through the ring 5′→3′ from the C- to the N-terminal side and thus is the correct polarity to serve as the translocating strand. We propose the grey, lagging strand DNA template will exit through the Mcm2/5 gate, possibly as a result of accumulation of ssDNA in the central channel (right). (**B**) A model for selecting the translocating strand during origin melting. Symmetric surfaces in different shades of green represent the two $MCM_N$ portions of a double hexamer. The dsDNA is first encircled by the MCM double hexamer (left panel). The dsDNA is driven toward the double hexamer interface by the dsDNA translocase activity of the AAA+ ATPase domains (not shown), which would be located above the light green surface and below the dark green surface. The dsDNA translocation creates strand separation where volume is available, enabling the MSSB to preferentially bind the strand with 5′→3′ clockwise polarity when viewed from the ATPase domain (middle panel). Importantly, the MSSB-bound strand corresponds to the strand upon which the MCM helicase will translocate during unwinding (right panel, magenta at top, cyan at bottom).

and Mcm5 are the only two subunits that lack a conserved positive residue at the *Pf*MCM R124 position, reducing ssDNA affinity and potentially facilitating strand exit. Following strand exit and extrusion of additional lengths of ssDNA, ring closure would poise each isolated hexamer to unwind DNA using a strand exclusion mechanism (*Fu et al., 2011*). The event that would drive double hexamer separation is unclear but could be facilitated by the change from encircling dsDNA to ssDNA, binding of additional factors (e.g., Mcm10) or modification of the helicase. A definitive test for this model awaits the development of assays that directly monitor the events of origin DNA melting and strand exclusion. Nevertheless, our studies provide structural and biochemical evidence that the MSSB is a critical ssDNA binding domain that functions during helicase loading and activation and provide initial insights into how ssDNA binding by MCM complex could facilitate selection of one strand during helicase activation.

## Materials and methods

### Cloning, mutagenesis, expression, and purification

An N-terminal His₆-SUMO-*Pf*MCM$_{1-256}$ expression construct was prepared. The original SUMO vector was the generous gift of Dr Christopher D Lima (*Mossessova and Lima, 2000*). An existing His₆-SUMO-tagged-fusion protein expression construct in a pRSFduet (EMD Millipore, Darmstadt, Germany) plasmid was treated with BamHI and XhoI to completely excise the original fusion partner to generate a BamHI site in-frame with the SUMO tag. This digested species was treated with phosphatase and gel-purified. A DNA fragment encoding the first 256 amino acids of *Pyrococcus furiosus* MCM was amplified by PCR with primers flanked by BamHI and SalI restriction sites. This fragment was digested with BamHI/SalI, ligated into the BamHI/XhoI-prepared vector, and was transformed into DH5α cells. The integrity of a single colony clone was verified by restriction digest pattern and by DNA sequencing (pLE009.3). Mutants were prepared by site-directed mutagenesis, and the sequences were verified by the Hartwell Center DNA Sequencing Facility (St. Jude Children's Research Hospital).

Expression plasmid pLE009.3 (WT), pCF001.1 (R124A), pCF002.1 (K129A), pCF003.1 (R186A), pCF004.1 (F202A), pCF0027.1 (K233A), or pCF009.1 (R124A/R186A) was transformed into BL21(DE3)-RIPL (Agilent Technologies, Santa Clara, CA) chemically competent cells and grown overnight in a 100 ml starter culture containing 30 mg/l kanamycin. The starter culture was distributed among 6 l of LB media containing 0.4% glucose and 30 mg/l kanamycin and grown to an O.D. of 0.3 at 37°C when the temperature was lowered to 18°C. When the O.D. had reached 0.7, expression was induced

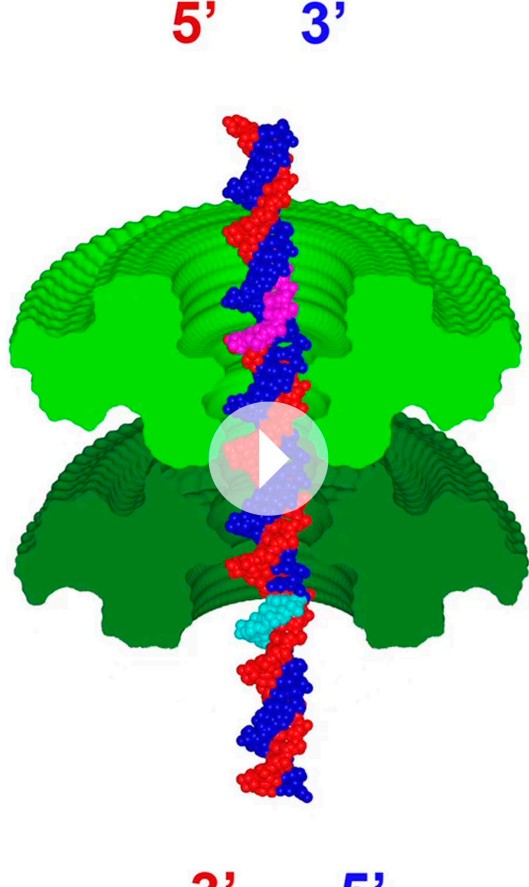

**Video 2**. Animation of a model for MCM to select the translocating strand during origin melting. Symmetric surfaces in different shades of green represent the two MCM$_N$ portions of a double hexamer. The dsDNA is first encircled by the MCM double hexamer. The dsDNA is driven toward the double hexamer interface by the dsDNA translocase activity of the AAA+ ATPase domains (not shown), which would be located above the light green surface and below the dark green surface. The dsDNA translocation creates strand separation where volume is available, enabling the MSSB to preferentially bind the strand with 5′→3′ clockwise polarity when viewed from the ATPase domain. Importantly, the MSSB-bound strand corresponds to the strand upon which the MCM helicase will translocate (magenta at top, cyan at bottom), as shown in *Figure 7B*, right panel.

by 0.5 mM IPTG, and the cells were grown for 16 hr at 18°C and harvested by centrifugation. The cells were lysed with a microfluidizer, and the soluble fraction was isolated by centrifugation and ammonium sulfate was added to 70% saturation. The precipitate was isolated by centrifugation, resuspended, and purified by Ni-NTA (Qiagen, Venlo, the Netherlands) chromatography. The elution was further purified by anion exchange, and the SUMO tag was removed by overnight digestion with Ulp1 protease (the Ulp1 protease plasmid was the generous gift of Dr Christopher D Lima, *Mossessova and Lima, 2000*). The NaCl concentration was raised to 1M, and the sample was passed over Ni-NTA resin, and the flowthrough was purified by anion exchange followed by gel filtration chromatography. The protein elutes at a volume consistent with a monomer. Pooled fractions were concentrated to 10–20 mg/ml. SDS-PAGE was used to assess the purity, and the protein concentration was determined by $A_{280}$ measurements ($\varepsilon$ = 11,460 M$^{-1}$ cm$^{-1}$ as determined by the ExPASy ProtParam tool). Purified *Pf*MCM$_N$ variants were stored at 4°C in buffer containing 20 mM HEPES, pH 7.6, 200 mM NaCl, 5 mM β-mercaptoethanol.

## Crystallization, data-collection, structure-solution, and refinement

Crystals of *Pf*MCM$_N$ in complex with a 30-mer poly-dT oligonucleotide were grown at 18°C in a hanging drop containing 1 µl of protein solution pre-mixed with a 30-mer poly-dT oligonucleotide (13.2 mg/ml protein; 120 µM poly-dT) and 2 µl of well solution (50 mM MES, pH 6.0, 10 mM Mg(OAc)$_2$, 28.5% PEG 3350). Data were collected at SER-CAT beamline 22-ID at the Advanced Photon Source at Argonne National Lab. Data were collected at 1.0 Å wavelength in 0.5° oscillations for a total of 190° of crystal rotation at 100 K. Data were integrated and scaled with the HKL-2000 package (*Otwinowski and Minor, 1997*) to 3.2 Å resolution. Initial phases were determined by molecular replacement by the program Phaser (*McCoy et al., 2007*) that placed 12 copies of a monomer of *Pf*MCM$_N$ (see below) in two hexamers. Following this placement, difference maps revealed strong electron density within the hexameric channels of both hexamers. The protein model was iteratively refined and manually improved until advancement ceased. At this stage, the difference electron density within the channel was observed at the 5-sigma level (*Figure 1—figure supplement 2*), and it was assigned as single-stranded DNA. The model was refined at various stages with CNS (*Brunger et al., 1998*; *Brunger, 2007*), phenix (*Afonine et al., 2012*), and refmac5 (*Vagin et al., 2004*). The final refinement was carried out with refmac5 using 3 TLS (*Winn et al., 2003*) groups for each protein monomer (one per subdomain). A Ramachandran plot calculated by Procheck (*Laskowski et al., 1993*) indicated the following statistics: core: 2244 (82.7%); allowed: 423 (15.6%); generously allowed: 48 (1.8%); disallowed: 0 (0%). Figures were

prepared with the program Bobscript (*Esnouf, 1997*) and rendered with the Raster3D (*Merritt and Bacon, 1997*) package or prepared with the program PyMOL (*Schrodinger, 2010*).

Crystals of $Pf$MCM$_N$ without DNA were grown at 18°C in a sitting drop containing 200 nl of protein solution (10 mg/ml) and 200 nl of well solution (0.2 M sodium malonate, pH 7.0, 20% PEG 3350). A plate crystal was cryoprotected by quickly passing it through well solution containing 15% ethylene glycol and flash frozen in liquid nitrogen. Data were collected at SER-CAT beamline 22-ID at the Advanced Photon Source at Argonne National Lab. Data were collected at 1.0 Å wavelength in 0.5° oscillations with two different segments of the same crystal. A total of 450 images were integrated and scaled with the HKL-2000 package (*Otwinowski and Minor, 1997*) to 2.65 Å resolution. The unit cell parameters are very close to hexagonal, but initial data merging showed the presence of a crystallographic twofold axis and a clear absence of a crystallographic threefold axis, indicating a monoclinic lattice. Initial phases were determined by molecular replacement by the program Molrep (*Vagin and Teplyakov, 1997*) by including a locked rotation and pseudo-translation. The program placed 6 copies of a monomer of $Mt$MCM$_N$ (*Fletcher et al., 2003*) as a single hexamer in the asymmetric unit in space group P2$_1$. The hexamers pack in layers with the hexameric axes mutually aligned parallel to the crystallographic 2$_1$ axis. Individual layers are highly sixfold symmetric, but a crystallographic 6-fold symmetry is precluded because the NCS 6-fold axes of successive layers are not mutually compatible. The model was refined at various stages with CNS (*Brunger et al., 1998*; *Brunger, 2007*), phenix (*Afonine et al., 2012*), and refmac5 (*Vagin et al., 2004*). The final refinement was carried out with refmac5 using 3 TLS (*Winn et al., 2003*) groups for each protein monomer (one per subdomain). A Ramachandran plot calculated by Procheck (*Laskowski et al., 1993*) indicated the following statistics: core: 1168 (85.8%); allowed: 183 (13.4%); generously allowed: 11 (0.8%); disallowed: 0 (0%). Figures were prepared with the program Bobscript (*Esnouf, 1997*) and rendered with the Raster3D (*Merritt and Bacon, 1997*) package.

## Electromobility shift assay

DNA-binding reactions were set up in 20 µl with varying concentrations of $Pf$MCM$_N$ (0–54 µM) and 160 nM 5′-fluorescein-labeled T40 ssDNA (Sigma-Aldrich, St. Louis, MO) in 20 mM HEPES, pH 7.6, 200 mM NaCl, 5 mM MgCl$_2$, and 5 mM βME. Reactions were incubated at 25°C in a BioRad DNA Engine thermocycler for 30 min. Loading buffer (2.5 mg/ml bromophenol blue and 40% sucrose; 5 µl) was added, and 5 µl were loaded in a 4–20% 1X TBE gradient PAGE gel (BioRad, Berkeley, CA) and run at 100 V for 105 min. Gels were imaged by a Fuji LAS-4000 with an 8 s exposure and a SYBR-Green filter. The fluorescence intensities of bands for the free and bound species were quantified with MultiGauge (GE Healthcare, Piscataway, NJ) and fit to two simultaneous equations with Prism (GraphPad Software, La Jolla, CA):

$$I(\text{free})/I_0 = K_{half}^h \Big/ \left(K_{half}^h + [\text{MCM}_N]^h\right) \; ; \; I(\text{bound})/I_0 = [\text{MCM}_N]^h \Big/ \left(K_{half}^h + [\text{MCM}_N]^h\right)$$

to determine the concentration of half-binding (K$_{half}$) and a hill coefficient (h). The dsDNA EMSAs were identical except that they included a 26-mer dsDNA substrate and a different concentration range of $Pf$MCM$_N$ (0–20 µM). The dsDNA substrate was prepared by annealing two oligos (5′-[Fluorescein]-ATGGCAGATCTCAATTGGATATCGGC-3′ and 5′-GCCGATATCCAATTGAGATCTGCCAT-3′, Sigma-Aldrich) followed by purification on a gel filtration column (GE Healthcare Superose 12 10/300).

## Yeast protein purification

Mcm2-7/Cdt1, Mcm4D6D/Cdt1, Mcm4D7D/Cdt1, Mcm6D7D/Cdt1 and Mcm4D6D7D/Cdt1 complexes were purified from 2 L cultures of ySKM01, ySKM02, ySKM03, ySKM04 and ySKM05, respectively. Cultures were grown to O.D. = 0.8 and arrested at G1 phase by addition of alpha factor (200 ng/ml) for two hours followed by induction of Mcm2-7/Cdt1 expression by addition of galactose to 2% for 4 hr. Harvested cell pellets were re-suspended in 1/3 pellet volume of cell lysis buffer (100 mM HEPES-KOH (pH 7.6), 1.5 M potassium glutamate, 0.8 M sorbitol, 10 mM magnesium acetate, 1 mM dithiothreitol and 1X Complete Protease Inhibitor Cocktail [Roche Diagnostics, Indianapolis, IN]) and frozen in liquid nitrogen. The frozen cell pellets were broken using a SPEX SamplePrep Freezer/Mill. After thawing, 15 ml of Buffer H (25 mM HEPES-KOH (pH 7.6), 1 mM EDTA, 1 mM EGTA, 5 mM magnesium acetate, 10% glycerol, 0.02% NP40) containing 0.5 M potassium glutamate, 3 mM ATP and 1X Complete Protease Inhibitor Cocktail was added to the broken cells. The cell lysate was centrifuged at 45,000×$g$ rpm for 90 min (Ti70 Rotor, Beckman) and the supernatant was mixed with 0.6 ml anti-Flag Agarose (Sigma-Aldrich) equilibrated with Buffer H containing 0.5 M potassium glutamate. The mix was incubated for 4 hr at 4°C. The resin was washed and Mcm2-7/Cdt1 complexes were

eluted with Buffer H containing 0.3 M potassium glutamate, 3 mM ATP and 0.15 mg/ml 3xFlag peptides. The eluted fractions were concentrated using Vivaspin 6 (Mw. cutoff 100 KDa, Sartorius) to 500 µl and applied to Superdex 200 HR 10/30 gel filtration column (GE Healthcare). For each mutant complex, the corresponding wild-type proteins were epitope-tagged with V5 (e.g., in the strain expressing the Mcm4D7D/Cdt1 the wild-type MCM4 and MCM7 genes were tagged with V5). This allowed the endogenous V5-tagged Mcm4, 6 or 7 subunits to be depleted by incubating with anti-V5 agarose (Sigma) before applying the MSSB mutant complexes to the gel filtration column.

## Helicase loading assay

2 pmole ORC, 3 pmole Cdc6 and 6 pmole Mcm2-7/Cdt1 were sequentially added to the 40 µl reaction solution containing 1 pmole of bead-coupled 1.3 Kbps *ARS1* DNA in helicase loading buffer (25 mM HEPES-KOH (pH7.6), 12.5 mM magnesium acetate, 0.1 mM zinc acetate, 300 mM potassium glutamate, 20 µM creatine phosphate, 0.02% NP40, 10% glycerol, 3 mM ATP, 1 mM dithiothreitol and 2 µg creatine kinase). The reaction mix was incubated at 25°C at 1200 rpm for 30 min in a thermomixer (Eppendorf). Beads were washed three times with Buffer H containing 0.3 M potassium glutamate and DNA bound proteins were eluted from the beads using DNase I. Eluted proteins were separated by SDS-PAGE and stained with a fluorescent protein stain (Krypton, Thermo Scientific). For high salt wash experiments, Buffer H containing 0.5 M NaCl was used at the second wash step. In ATPγS experiments, 6 mM ATPγS was used instead of ATP.

## In vitro replication assay

Helicase loading reactions were performed using 0.5 pmole ORC, 0.75 pmole Cdc6 and 2 pmole MCM/Cdt1 and 250 fmole bead-coupled 3.6 Kbps circular pUC19-*ARS1* plasmid DNA (*Heller et al., 2011*). Origin-loaded MCM complexes were phosphorylated with 450 µg purified DDK in DDK reaction buffer (50 mM HEPES-KOH (pH7.6), 3.5 mM magnesium acetate, 0.1 mM zinc acetate, 150 mM potassium glutamate, 0.02% NP40, 10% glycerol, 1 mM spermine, 1 mM ATP and 1 mM dithiothreitol, 30 µl). Phosphorylated MCM complexes were then activated with 750 µg S phase extract in the replication reaction buffer (25 mM HEPES-KOH (pH7.6), 12.5 mM magnesium acetate, 0.1 mM zinc acetate, 300 mM potassium glutamate, 20 µM creatine phosphate, 0.02% NP40, 10% Glycerol, 3 mM ATP, 40 µM dNTPs, 200 µM CTP/UTP/GTP, 1 mM dithiothreitol, 10 µCi [α-P$^{32}$] dCTP and 2 µg creatine kinase, 40 µl) for 1 hr at 25°C and 1200 RPM in a Thermomixer (Eppendorf). After the reaction, DNA synthesis was monitored using alkaline agarose gel. DNA bound proteins were released from the beads by DNase I treatment and analyzed by immunoblot. S phase extracts were prepared from ySKS10 and ySKS11 as described previously (*Heller et al., 2011*).

## S. cerevisiae in vivo complementation assay

MSSB mutations were introduced into TRP + ARS/CEN plasmids containing *MCM4*, *MCM6*, or *MCM7* under the control of the MCM5 promoter. The resultant constructs were tested for *MCM4*, *MCM6*, or *MCM7* function by a plasmid shuffle assay (*Schwacha and Bell, 2001*). To test the double mutant complementation, one MSSB mutant Mcm subunit (either *MCM4* or *MCM6*) was integrated into a plasmid shuffle strain for a second subunit.

## Strain construction for in vivo complementation assay

To integrate MSSB mutations into the chromosomal locus, we constructed plasmids containing the *MCM4* or *MCM6* promoter upstream of a NatMX4 (for *MCM4*) or *LEU2* (for *MCM6*) marker cassette, with the Mcm5 promoter plus the *MCM4* or *MCM6* gene downstream of the marker and restriction enzyme sites flanking the entire integration unit (pSKC04 and pSKC05, respectively). Proper integration was confirmed by PCR followed by sequencing.

To create strains carrying MSSB mutations in *MCM4* and *MCM6* or *MCM6* and *MCM7*, we began with strains carrying mcm4 or mcm7 deletion and the wild-type copy of *MCM4* or *MCM7* on URA+ ARS/CEN constructs, respectively. MCM6 MSSB mutation was integrated into these strains using the LEU+ integrating construct described above. For a strain carrying MSSB mutations in *MCM4* and *MCM7*, *MCM4* MSSB mutations were incorporated in to a strain carrying mcm7 deletion and wild-type copy of *MCM7* on URA+ ARS/CEN constructs, using NAT+ integrating construct. Then TRP+ ARS/CEN plasmids carrying *MCM4* or *MCM7* MSSB mutant allele were transformed above strains. Proliferation of double-mutant strains was analyzed using FOA counter-selection against the URA+ wild-type *MCM4* or *MCM7* plasmid.

Yeast strains and plasmids of this study are listed in *Tables 2 and 3*.

Biochemistry | Biophysics and structural biology

**Table 2.** Yeast strains used in this study

| Strains | Genotype | Source |
|---|---|---|
| ySKM01 | *ade2-1 trp1-1 leu2-3,112 his3-11,15 ura3-1 can1-100 bar1::HisG lys2::HisG pep4Δ::unmarked* | This study |
| | *his3::pSKM004 (GAL1,10-MCM2, Flag-MCM3) leu::pSKM007 (GAL1, 10-Cdt1-Strep, GAL4)* | |
| | *lys::pSKM002 (GAL1,10-MCM4, MCM5)* | |
| | *trp::pSKM003 (GAL1,10-MCM6, MCM7)* | |
| ySKM02 | *ade2-1 trp1-1 leu2-3,112 his3-11,15 ura3-1 can1-100 bar1::HisG lys2::HisG pep4Δ::KanMX6* | This study |
| | *MCM4-V5 (NatMX4) MCM6-V5 (CaURA3MX4) MCM7-V5 (HphMX4)* | |
| | *his3::pSKM004 (GAL1,10-MCM2, Flag-MCM3) leu::pSKM007 (GAL1, 10-Cdt1-Strep, GAL4)* | |
| | *lys::pSKM008 (GAL1,10-mcm4[R334A/K398A], MCM5) trp::pSKM009 (GAL1,10-mcm6[R296A/R360A], MCM7)* | |
| ySKM03 | *ade2-1 trp1-1 leu2-3,112 his3-11,15 ura3-1 can1-100 bar1::HisG lys2::HisG pep4Δ::KanMX6* | This study |
| | *MCM4-V5 (NatMX4) MCM6-V5 (CaURA3MX4) MCM7-V5 (HphMX4)* | |
| | *his3::pSKM004 (GAL1,10-MCM2, Flag-MCM3) leu::pSKM007 (GAL1, 10-Cdt1-Strep, GAL4)* | |
| | *lys::pSKM008 (GAL1,10-mcm4[R334A/K398A], MCM5) trp::pSKM010 (GAL1,10-MCM6, mcm7[R247A/K314A])* | |
| ySKM04 | *ade2-1 trp1-1 leu2-3,112 his3-11,15 ura3-1 can1-100 bar1::HisG lys2::HisG pep4Δ::KanMX6* | This study |
| | *MCM4-V5 (NatMX4) MCM6-V5 (CaURA3MX4) MCM7-V5 (HphMX4)* | |
| | *his3::pSKM004 (GAL1,10-MCM2, Flag-MCM3) leu::pSKM007 (GAL1, 10-Cdt1-Strep, GAL4)* | |
| | *lys::pSKM002 (GAL1,10-MCM4, MCM5)* | |
| | *trp::pSKM011 (GAL1,10-mcm6[R296A/R360A], mcm7[R247A/K314A])* | |
| ySKM05 | *ade2-1 trp1-1 leu2-3,112 his3-11,15 ura3-1 can1-100 bar1::HisG lys2::HisG pep4Δ::KanMX6* | This study |
| | *MCM4-V5 (NatMX4) MCM6-V5 (CaURA3MX4) MCM7-V5 (HphMX4)* | |
| | *his3::pSKM004 (GAL1,10-MCM2, Flag-MCM3) leu::pSKM007 (GAL1, 10-Cdt1-Strep, GAL4)* | |
| | *lys::pSKM008 (GAL1,10-mcm4[R334A/K398A], MCM5)* | |
| | *trp::pSKM011 (GAL1,10-mcm6[R296A/R360A], mcm7[R247A/K314A])* | |
| ySKS10 | *ade2-1 trp1-1 leu2-3,112 his3-11,15 ura3-1 can1-100 lys2::HisG pep4Δ::Hph cdc7-4* | This study |
| | *pol1-5xFlag (KanMX4)* | |
| | *leu::pSKS001 (GAL1,10-Cdc45-V5, Sld3-S)* | |
| | *lys::pSKS002 (GAL1,10-Dpb11-VSVG, Sld2-HSV)* | |
| | *ura::pSKS003 (Gal1,10-Cdc28, Clb5)* | |
| | *his::pSKS004 (Gal1,10-Sld7)* | |
| ySKS11 | *ade2-1 trp1-1 leu2-3,112 his3-11,15 ura3-1 can1-100 lys2::HisG pep4Δ::Hph cdc7-4* | This study |
| | *pol2-5xFlag (KanMX4)* | |
| | *leu::pSKS001 (GAL1,10-Cdc45-V5, Sld3-S)* | |
| | *lys::pSKS002 (GAL1,10-Dpb11-VSVG, Sld2-HSV)* | |
| | *ura::pSKS003 (Gal1,10-Cdc28, Clb5)* | |
| | *his::pSKS004 (Gal1,10-Sld7)* | |
| ASY1059.1 | *MatA, ade2-1, ura3-11, his3-11,15, leu2-3,12, can-100, trp1-1* | (*Schwacha and Bell, 2001*) |
| | *mcm4 Δ::hisG/pAS412 (ARS/CEN URA+ PMCM5-MCM4-HA/HIS)* | |

*Table 2. Continued on next page*

*Table 2. Continued*

| Strains | Genotype | Source |
|---|---|---|
| ASY2157 | **MatA, ade2-1, ura3-11, his3-11,15, leu2-3,12, can-100, trp1-1, lys2::hisG, bar1::hisG, PEP4 Δ::KANMX4,** | (*Schwacha and Bell, 2001*) |
| | **MCM6 Δ::HISMX6/pAS452 (ARS/CEN URA+ PMCM5-MCM6-HA/HIS)** | |
| ASY1050.1 | **MatA, ade2-1, ura3-11, his3-11,15, leu2-3,12, can-100, trp1-1** | (*Schwacha and Bell, 2001*) |
| | **mcm7Δ::hisG/pGEMCDC47 (ARS/CEN URA+ MCM7)** | |
| ySKC01 | **MatA, ade2-1, ura3-11, his3-11,15, leu2-3,12, can-100, trp1-1** | This study |
| | **mcm4 Δ::hisG/pAS412 (ARS/CEN URA+ PMCM5-MCM4-HA/HIS) mcm6::LEU2-PMCM5-mcm6[R296A/R360A]** | |
| ySKC02 | **MatA, ade2-1, ura3-11, his3-11,15, leu2-3,12, can-100, trp1-1** | This study |
| | **mcm7Δ::hisG/pGEMCDC47 (ARS/CEN URA+ MCM7) mcm6::LEU2-PMCM5-mcm6[R296A/R360A]** | |
| ySKC03 | **MatA, ade2-1, ura3-11, his3-11,15, leu2-3,12, can-100, trp1-1** | This study |
| | **mcm7Δ::hisG/pGEMCDC47 (ARS/CEN URA+ MCM7) mcm4::NatMX4-PMCM5-mcm4[R334A/K398A]** | |

**Table 3.** Yeast plasmids used in this study

| Plasmids | Description | Source |
|---|---|---|
| pSKM002 | **pRS307 (GAL1,10-MCM4, MCM5)** | This study |
| pSKM003 | **pRS404 (GAL1,10-MCM6, MCM7)** | This study |
| pSKM004 | **pRS403 (GAL1,10-MCM2, Flag-MCM3)** | This study |
| pSKM007 | **pRS305 (GAL1,10-Cdt1-Strep, GAL4)** | This study |
| pSKM008 | **pRS307 (GAL1,10-mcm4[R334A/K398A], MCM5)** | This study |
| pSKM009 | **pRS404 (GAL1,10-mcm6[R296A/R360A], MCM7)** | This study |
| pSKM010 | **pRS404 (GAL1,10-MCM6, mcm7[R247A/K314A])** | This study |
| pSKM011 | **pRS404 (GAL1,10-mcm6[R296A/R360A], mcm7[R247A/K314A])** | This study |
| pSKS001 | **pRS305 (GAL1,10-Cdc45-V5, Sld3-S)** | This study |
| pSKS002 | **pRS307 (GAL1,10-Dpb11-VSVG, Sld2-HSV)** | This study |
| pSKS003 | **pRS306 (Gal1,10-Cdc28, Clb5)** | This study |
| pSKS004 | **pRS403 (Gal1,10-Sld7)** | This study |
| pSKC001 | **pRS414 (PMCM5-mcm4[R334A/K398A])** | This study |
| pSKC002 | **pRS414 (PMCM5- mcm6[R296A/R360A])** | This study |
| pSKC003 | **pRS414 (PMCM5-mcm7[R247A/K314A])** | This study |
| pSKC004 | **pRS414 (PMCM4-NatMX4-PMCM5- mcm4[R334A/K398A])** | This study |
| pSKC005 | **pRS414 (PMCM6-LEU2-PMCM5- mcm6[R296A/R360A])** | This study |

# Acknowledgements

Data were collected at Southeast Regional Collaborative Access Team (SER-CAT) 22-ID beamline at the Advanced Photon Source, Argonne National Laboratory. Supporting institutions may be found at www.ser-cat.org/members.html. We are grateful to SER-CAT staff for experimental support. Use of the Advanced Photon Source was supported by the U S Department of Energy, Office of Science, Office of Basic Energy Sciences, under Contract No. W-31-109-Eng-38. We thank Dr Amanda Nourse (Hartwell Center for Biotechnology and Bioinformatics, St Jude Children's Research Hospital) for sample analysis by analytical ultracentrifugation and Brett Waddell (Hartwell Center for Biotechnology and Bioinformatics, St Jude Children's Research Hospital) for preliminary DNA-binding experiments by surface plasmon resonance. We thank Dr Janet Partridge, Dr Brenda Schulman, Dr Stephen White, Dr Ishara Azmi and Simina Ticau for providing comments on the manuscript.

## Additional information

### Funding

| Funder | Grant reference number | Author |
|---|---|---|
| American Lebanese Syrian Associated Charities | | Eric J Enemark |
| National Institute of General Medical Sciences | R01GM098771 | Eric J Enemark |
| Cancer Center Support Grant, National Cancer Institute | 5 P30 CA021765-32 | Eric J Enemark |
| Howard Hughes Medical Institute | | Stephen P Bell |
| National Institute of General Medical Sciences | R01-GM52339 | Stephen P Bell |

The funders had no role in study design, data collection and interpretation, or the decision to submit the work for publication.

### Author contributions

CAF, SK, Acquisition of data, Analysis and interpretation of data, Drafting or revising the article; LBE, Acquisition of data, Contributed unpublished essential data or reagents; SPB, EJE, Conception and design, Analysis and interpretation of data, Drafting or revising the article

## Additional files

### Major datasets

The following datasets were generated:

| Author(s) | Year | Dataset title | Dataset ID and/or URL | Database, license, and accessibility information |
|---|---|---|---|---|
| Froelich CF, Kang S, Epling LB, Bell SP, Enemark EJ | 2014 | MCM N-terminal domain crystal structure without DNA | 4POF; http://www.rcsb.org/pdb/explore/explore.do?structureId=4POF | Publicly available at RCSB Protein Data Bank. |
| Froelich CF, Kang S, Epling LB, Bell SP, Enemark EJ | 2014 | MCM N-terminal domain: ssDNA co–crystal structure | 4POG; http://www.rcsb.org/pdb/explore/explore.do?structureId=4POG | Publicly available at RCSB Protein Data Bank. |

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
