## [Decision Letter]

Thank you for sending your work entitled “A conserved MCM single-stranded DNA binding element is essential for replication initiation” for consideration at *eLife*. Your article has been favorably evaluated by a Senior editor and 3 reviewers, one of whom, Michael Botchan, is a member of our Board of Reviewing Editors, and one of whom, James Berger, has agreed to reveal his identity.

The Reviewing editor and the other reviewers discussed their comments before we reached this decision, and the Reviewing editor has assembled the following comments to help you prepare a revised submission.

The data and discussion presented in this manuscript are important contributions to our present knowledge as to how chromosomal duplex DNA is unwound by the Mcm helicases. Furthermore the crystal structure from the Archeal Mcm from *Pf* MCM combined with the in vivo and biochemical studies from the eukaryote *Sc* MCM's allow for exciting speculation as to how the earliest steps in initiation might occur. The work allows for a plausible model for the mechanism of how the leading strand might be selected in the duplex DNA concomitant with the melting and extrusion of the lagging strand from the central channel. It should be accepted for publication by *eLife* with minor revisions. A few significant points are listed below.

1) The double hexamer loading reactions with the purified *Sc* proteins show that the amino terminal OB fold DNA-binding residues revealed by the x-ray structure do in fact play a role in the establishment of the PreRC in the eukaryote. It seems possible that the triple mutations (in subunits 4, 6 & 7) would be even more defective in that step than the double mutants. While we don't ask the authors to test this, perhaps a more balanced view would be that the amino terminal domain is important at all stages of the replication pathway as the DH transforms from an inactive structure to an active trans-locating structure. Rather than more important at one stage than another. This would be consistent with prior studies on Archeal Mcm's and the present study. It is hard to access how a fold drop in an in vitro reaction might translate to an in vivo complementation test. The OB fold DNA contacts may be key to the proper closure of the ring after first recruitment. Then again in the strand exclusion step that occurs simultaneously with tight GINS binding as the data here also show.

2) The discussion should also include the possibility that in full intact hexamers and double hexamers contacts in the MCM_N_ domain might be a bit different at each stage. This leads to another question. Will duplex or single strand binding activities for the intact hexamer of the *Pf* Mcm's involve the same or additional contacts through the N-terminus in conditions when non hydrolyzable ATP is in the buffer? While we are not suggesting that the surprising path of the leading stand in the amino-terminal domain of the Mcm's would change in the intact protein subtle switches might add additional complexity to the binding mode(s). This should be discussed or additional data presented.

3) A final point that needs to be addressed in a revised manuscript: the authors postulate that the MSSB binding mode might be specific to helicase loading and origin activation, while not necessarily involved in the elongation step of DNA replication (MCM motor translocation). To substantiate this statement they point out that the isolated AAA+ domain from archaeal MCMs is an active helicase, while the N-terminal domain is not strictly required for DNA unwinding. The authors should consider and cite the work from Stephen D. Bell's laboratory ([8], NAR, which they cite only in part), indicating that the isolated MCM AAA+ domain can indeed function as a helicase, but it contains a promiscuous activity, being able to unwind blunt duplex DNA as well as a primer-template junction containing either a 3' or a 5' tail. Addition of the MCM NTD in trans restores the 3'->5' polarity of DNA translocation for the MCM motor, as observed with the full-length enzyme. While S.D. Bell's biochemical experiments agree with the present study in indicating that the N-terminal domain has an important role in strand selection, they also show that strand selection is key to productive DNA unwinding. As a corollary, we believe that a possible role for the MSSB site during fork unwinding should be suggested with greater strength.

---

## [Author Response]

*1) The double hexamer loading reactions with the purified* Sc *proteins show that the amino terminal OB fold DNA-binding residues revealed by the x-ray structure do in fact play a role in the establishment of the PreRC in the eukaryote. It seems possible that the triple mutations (in subunits 4, 6 & 7) would be even more defective in that step than the double mutants. While we don't ask the authors to test this, perhaps a more balanced view would be that the amino terminal domain is important at all stages of the replication pathway as the DH transforms from an inactive structure to an active trans-locating structure. Rather than more important at one stage than another. This would be consistent with prior studies on Archeal Mcm's and the present study. It is hard to access how a fold drop in an* in vitro *reaction might translate to an* in vivo *complementation test. The OB fold DNA contacts may be key to the proper closure of the ring after first recruitment. Then again in the strand exclusion step that occurs simultaneously with tight GINS binding as the data here also show*.

We have analyzed recruitment and loading in the Mcm4/6/7 triple mutant. This mutant is indeed more defective in loading than any of the double mutants. We have added the triple mutant loading experiments to Figure 5—figure supplement 2 and discuss these in the helicase loading sections of the Results and Discussion. This mutant supports the idea that there is a role of the MSSB in either loading or retention of the Mcm2-7 complex on the DNA during initiation of replication. In performing these experiments we repeated the analysis of the double mutants, and we have added quantitation of these data to Figure 5—figure supplement 2.

The discussion is more neutral in differentiating the relative importance of the MSSB for loading, activation, and potentially during elongation. The Discussion has been reorganized to describe these three possibilities in three sequential paragraphs that follow the initial Discussion paragraph.

We have removed the first sentence from the last paragraph of the Results section on in vitro reaction expectation from complementation experiments (“Because the extent of helicase loading defects did not account for the lethal phenotypes of the mutations, we looked for additional defects in replication initiation”). We agree with the reviewers that we cannot be sure how in vitro data will correlate with in vivo phenotypes.

*2) The discussion should also include the possibility that in full intact hexamers and double hexamers contacts in the MCM*_*N*_
*domain might be a bit different at each stage. This leads to another question. Will duplex or single strand binding activities for the intact hexamer of the* Pf *Mcm's involve the same or additional contacts through the N-terminus in conditions when non hydrolyzable ATP is in the buffer? While we are not suggesting that the surprising path of the leading stand in the amino-terminal domain of the Mcm's would change in the intact protein subtle switches might add additional complexity to the binding mode(s). This should be discussed or additional data presented*.

Our structure indicates that subtle changes to the subunit:subunit configuration at the N-terminal domain strongly impact the ability of the MSSB to interact with ssDNA. A coupling of these subtle changes to the ATPase domains would provide a straightforward and attractive means for MCM to switch affinity or preference for different forms of DNA based upon ATP-binding, or even to repetitively change binding via the ATPase cycle.

We have added in the discussion on a possible role for the MSSB during unwinding that the ATPase domain could readily modulate the subunit:subunit configuration and thus change the behavior of MSSB:DNA binding, perhaps through a conserved “allosteric communication loop”, ACL, and we have added references for the ACL. While we envision that the presence of ATP at an associated ATPase domain could readily influence binding by the MSSB, we don’t have a basis to predict whether ATP binding would tend to increase or decrease binding to ssDNA by the MSSB. We have also added to this paragraph that the MSSB might bind ssDNA differently during unwinding, possibly binding in a mode more like the OB-fold protein SSB such that the ssDNA runs more parallel to the channel during that stage.

*3) A final point that needs to be addressed in a revised manuscript: the authors postulate that the MSSB binding mode might be specific to helicase loading and origin activation, while not necessarily involved in the elongation step of DNA replication (MCM motor translocation). To substantiate this statement they point out that the isolated AAA+ domain from archaeal MCMs is an active helicase, while the N-terminal domain is not strictly required for DNA unwinding. The authors should consider and cite the work from Stephen D. Bell's laboratory (*[8]*, NAR, which they cite only in part), indicating that the isolated MCM AAA+ domain can indeed function as a helicase, but it contains a promiscuous activity, being able to unwind blunt duplex DNA as well as a primer-template junction containing either a 3' or a 5' tail. Addition of the MCM NTD in trans restores the 3'->5' polarity of DNA translocation for the MCM motor, as observed with the full-length enzyme. While S.D. Bell's biochemical experiments agree with the present study in indicating that the N-terminal domain has an important role in strand selection, they also show that strand selection is key to productive DNA unwinding. As a corollary, we believe that a possible role for the MSSB site during fork unwinding should be suggested with greater strength*.

The Discussion paragraph has been revised to include the increased processivity of the Sso MCM helicase when the N-terminal domain is present (8), and that the MSSB could contribute to enhanced processivity during unwinding. The paragraph also suggests the possibility that during unwinding, the MSSB:DNA interactions could change to a configuration that resembles that of the OB-fold SSB, which would orient the ssDNA along the central channel rather than perpendicular to it. The previously observed specificity of polarity derived from the N-terminal domain (8) has been added at the conclusion of the subsequent paragraph about strand selection during the transition from binding dsDNA to ssDNA.